# Individual and community factors determining delayed leprosy case detection: A systematic review

**Yudhy Dharmawan** [1,2] *, **Ahmad Fuady** [1,3], **Ida Korfage** [1], **Jan Hendrik Richardus** [1]

**1** Department of Public Health, Erasmus MC, University Medical Center Rotterdam, Rotterdam, The Netherlands, **2** Faculty of Public Health, Universitas Diponegoro, Semarang, Indonesia, **3** Department of Community Medicine, Faculty of Medicine, Universitas Indonesia, Jakarta, Indonesia

* y.dharmawan@erasmusmc.nl, yudhydharmawan@lecturer.undip.ac.id (YD)

## Abstract

### Background

The number of new leprosy cases is declining globally, but the disability caused by leprosy remains an important disease burden. The chance of disability is increased by delayed case detection. This review focusses on the individual and community determinants of delayed leprosy case detection.

### Methods

This study was conducted according to the PRISMA guidelines (Preferred Reporting Items for Systematic Reviews and Meta-Analysis). The study protocol is registered in PROS-PERO (code: CRD42020189274). To identify determinants of delayed detection, data was collected from five electronic databases: Embase.com, Medline All Ovid, Web of Science, Cochrane CENTRAL, and the WHO Global Health Library.

### Results

We included 27 papers from 4315 records assessed. They originated in twelve countries, had been published between January 1, 2000, and January 31, 2021, and described the factors related to delayed leprosy case detection, the duration of the delayed case, and the percentage of Grade 2 Disability (G2D). The median delay in detection ranged from 12 to 36 months, the mean delay ranged from 11.5 to 64.1 months, and the percentage of G2D ranged from 5.6 to 43.2%. Health-service-seeking behavior was the most common factor associated with delayed detection. The most common individual factors were older age, being male, having a lower disease-symptom perception, having multibacillary leprosy, and lack of knowledge. The most common socioeconomic factors were living in a rural area, performing agricultural labor, and being unemployed. Stigma was the most common social and community factor.

**Data Availability Statement:** All relevant data are within the manuscript and its Supporting Information files.

**Funding:** This work was done as part of a Ph.D. scholarship in Health Science at Erasmus MC, University Medical Center Rotterdam, generously provided by Universitas Diponegoro in Indonesia to YD. The funder of this scholarship played no role in the study design, data collection and analysis, decision to publish, or preparation of the manuscript.

**Competing interests:** The authors have declared that no competing interests exist.

## Conclusions

Delayed leprosy case detection is clearly correlated with increased disability and should therefore be a priority of leprosy programs. Interventions should focus on determinants of delayed case detection such as health-service-seeking behavior, and should consider relevant individual, socioeconomic, and community factors, including stigmatization. Further study is required of the health service-related factors contributing to delay.

### Author summary

Leprosy remains an important public health problem with many new leprosy patients diagnosed with visible physical deformities, indicating a long delay in the detection of cases. For effective prevention programs, it is important to know the factors at the level of the individual and the community that contribute to the delay. We reviewed all published studies that reported individual and community factors related to delayed case detection in leprosy and included 27 studies in our analysis, published between January 1, 2000, and January 31, 2021. Health-service-seeking behavior was the most common factor associated with delay in case detection. The most common individual factors were older age, being male, having a lower disease-symptom perception, having multibacillary leprosy, and lack of knowledge about leprosy. The most common socioeconomic factors were living in a rural area, performing agricultural labor, and being unemployed. Stigma was the most common social and community factor associated with detection delay.

The presence of physical disability in newly diagnosed leprosy patients is clearly related to the delay in detecting these patients. Leprosy control interventions should take factors related to detection delay into account more comprehensively. Also, there is a need to study health service-related factors that contribute to detection delay of leprosy patients.

## Introduction

Although leprosy is caused by *Mycobacterium leprae*, only a small percentage of those infected with this microorganism develop clinical disease. *M. leprae* is slow-growing and has an incubation period ranging from 2 to 12 years. While the mode of transmission has not been established conclusively, person-to-person spread via nasal droplets is believed to be the main route [1].

Due to the irreversible disability and the social stigma it causes, leprosy has been a public health problem for many centuries. Fortunately, leprosy control has improved markedly over the past decades, with the leprosy annual new case detection falling from around 750,000 in 2000 to just over 200,000 in 2019 [2]. This decline occurred after the world-wide introduction of multidrug therapy (MDT) in the 1980s, which was combined with nationwide health education, case-finding campaigns, and improvements in the quality of leprosy treatment by health services in endemic countries [3]. Between them, India, Brazil, and Indonesia currently account for 80% of the new cases detected worldwide [2].

Another important indicator of the burden of disease beside incidence rate is the number of new cases with Grade 2 Disability (G2D), which are defined as people with leprosy who have visible deformities due to leprosy neuropathy [4]. Although the worldwide percentage of new cases with G2D fell slightly from 5.8% in 2010 to 5.3% in 2019, the percentage of new cases with G2D was higher in 2019 in Brazil (8.4%), Ethiopia (12.8%), Nepal (6.6%), and

Nigeria (15.2%) than in 2018 (7.4%, 8.0%, 4.1%, 14.6%, respectively) [2]. In 2019, India reported 2761 new cases with G2D, Brazil 2351, and Indonesia 1121.

The disability caused by leprosy remains an important disease burden. The target stated in the WHO Global Leprosy Strategy 2016–2020 –i.e. less than one newly diagnosed G2D leprosy case per million population in 2020 [5]–was not achieved.

G2D has been proposed as a more appropriate indicator for disease burden than leprosy prevalence (defined as the number of patients receiving treatment at the end of a calendar year): it is less susceptible to operational factors such as quality of control programs, and is also a more robust marker for mapping cases of leprosy per country [6]. New G2D cases are also an indicator of delayed leprosy detection [7]. The transmission of *M. leprae* is augmented by delays in detection, diagnosis, and treatment, all of which may also lead to progression of the disease in terms of increased nerve impairment, sensory loss, and the resulting disability [8]. Indirectly, G2D also provides information on other factors that influence case detection, such as community awareness about leprosy, the capacity of health staff to recognize early signs and symptoms, and, to some extent, the quality of the leprosy health services themselves [9]. For these reasons, the WHO's strategy for reducing delays in case detection gives precedence to interventions that can detect cases before visible deformities occur [5].

To reduce delays in leprosy case detection, it is necessary to identify their individual and community determinants; this will support the planning and implementation of appropriate public health interventions. This systematic review is therefore intended to identify the determinants in question.

## Methods

This systematic review complies with the PRISMA guidelines (Preferred Reporting Items for Systematic Reviews and Meta-Analysis) [10]. The study protocol is registered in PROSPERO with reference code CRD42020189274.

### Selection criteria and search

In this systematic review, we searched for delayed leprosy case detection based on (a) the period of delay calculated from the beginning of signs or symptoms to diagnosis, either in numerical or categorical values; and (b) the occurrence of Grade 2 Disability (G2D). To identify factors determining delayed detection, we performed a systematic search of five electronic databases: Embase, Medline All Ovid, Web of Science, Cochrane CENTRAL, and the WHO Global Health Library (see S1 Text for details of the search strategy). We included leprosy-related original empirical studies that had been published in English between January 1, 2000, and January 31, 2021. We excluded case reports, articles without full text (abstract only), and articles that mentioned neither delayed case detection nor factors associated with delayed case detection.

To select articles for full-text screening, two reviewers (YD and AF) independently screened article titles and abstracts. Data from articles were extracted and double-entered into Microsoft Excel. Disagreements were settled by a third reviewer (IK or JHR). The extracted data included author(s), year of publication, article title, journal title, study design, study setting, number of study participants, type of measurement of delayed case finding (duration of the delay or presence of G2D), length of delay (in months or years), percentage of G2D, and data on correlations between leprosy delayed case finding and disabilities. We finally summarized factors related with delayed case detection in four sections: health-service-seeking behavior, individual factors, socioeconomic factors, and social and community factors. The factors were expressed as Odds Ratios (ORs), adjusted Odds Ratios (aORs), Hazard Ratios (HRs), and/or significance

(P) values. Methods and results are reported following the PRISMA guidelines (see S1 Table for the description).

## Evaluation of the quality of studies

The quality of articles was assessed using a risk-of-bias instrument for potential biases regarding study design. For quantitative studies, we used a scoring checklist to assess the quality of the research hypothesis; to assess the study population, selection bias, exposure, outcome, confounding; and also to formulate an overall opinion of the study's validity and applicability [11]. For qualitative studies, we used a COREQ checklist to evaluate research team and reflexivity; study design; and analysis and findings [11,12]. For mixed-method studies, both methods were combined. Quality was evaluated by two reviewers (YD and AF). In cases of disagreement, a third reviewer (IK or JHR) was invited to resolve the issue.

## Results

Through a systematic search, we identified 7048 studies that had been published in five databases between January 2000 and January 2021. During data extraction, two further papers were identified by the snowball method. After removing duplicates, 4315 studies remained; after title and abstract screening, 67 full articles were assessed for eligibility. In the final stage, we included 27 studies for analysis. Fig 1 shows the flowchart of article selection according to the PRISMA guidelines.

Nineteen studies were observational studies with quantitative analysis [13–31]. Fourteen of these were cross-sectional studies [15–21,24,25,27–31], one was a case-control study [13], two had a longitudinal cohort design [14,26] and two had retrospective analysis [22,23]. Six studies used mixed-methods analysis [32–37], and two used qualitative analysis [38,39]. Almost two-thirds of the studies (n = 17, 63%) collected data through interviews [13,15,16,21,23–25,28,30,32–39]; while nine studies assessed delayed case detection by reviewing medical records [14,17–19,22,26,29,31]; and one assessed delayed case detection through a self-administered questionnaire [27].

Seventeen studies had been conducted in Asia, with six in India, four in China, three in Nepal, two in Bangladesh, one each in Iran and Myanmar. Eight studies had been conducted in South America, with five in Brazil and one each in Colombia, Peru, and Paraguay. One study had been conducted in Africa (Ethiopia) and one in Europe (United Kingdom).

Studies had been conducted in various settings: community (n = 10), hospital (9), clinic (4), mixed hospital and clinic (1), mixed community and clinic (1), mixed clinic and a region aggregate data (1) and a nation-wide data assessment (1). As well as assessing the experiences of leprosy patients, studies had also involved health-care professionals [36–38], pastors [38] and parents of leprosy patients [32]. Detailed information on the selected studies is given in Table 1.

### Detection delay and Grade 2 Disability

Leprosy case detection delay was reported in various ways. Fifteen studies reported the delay in months or years (median, mean, or both). The median values (as reported in 12 studies) ranged from 12 to 36 months, while the mean values (as reported in 14 studies) ranged from 11.5 to 64.1 months (Fig 2). Five studies reported the delay in terms of categorical values. Delay was reported in weeks (e.g., 0–2 weeks); in months (e.g., 1–3 months, 3–6 months, etc.); or in years (e.g., 1–2 years, 3–5 years, etc.). One paper distinguished between patient delay and health-system delay [27]. One reported delay for adults and children separately [33]. Sixteen studies reported delayed case detection as percentage of G2D, which ranged from 5.6% to

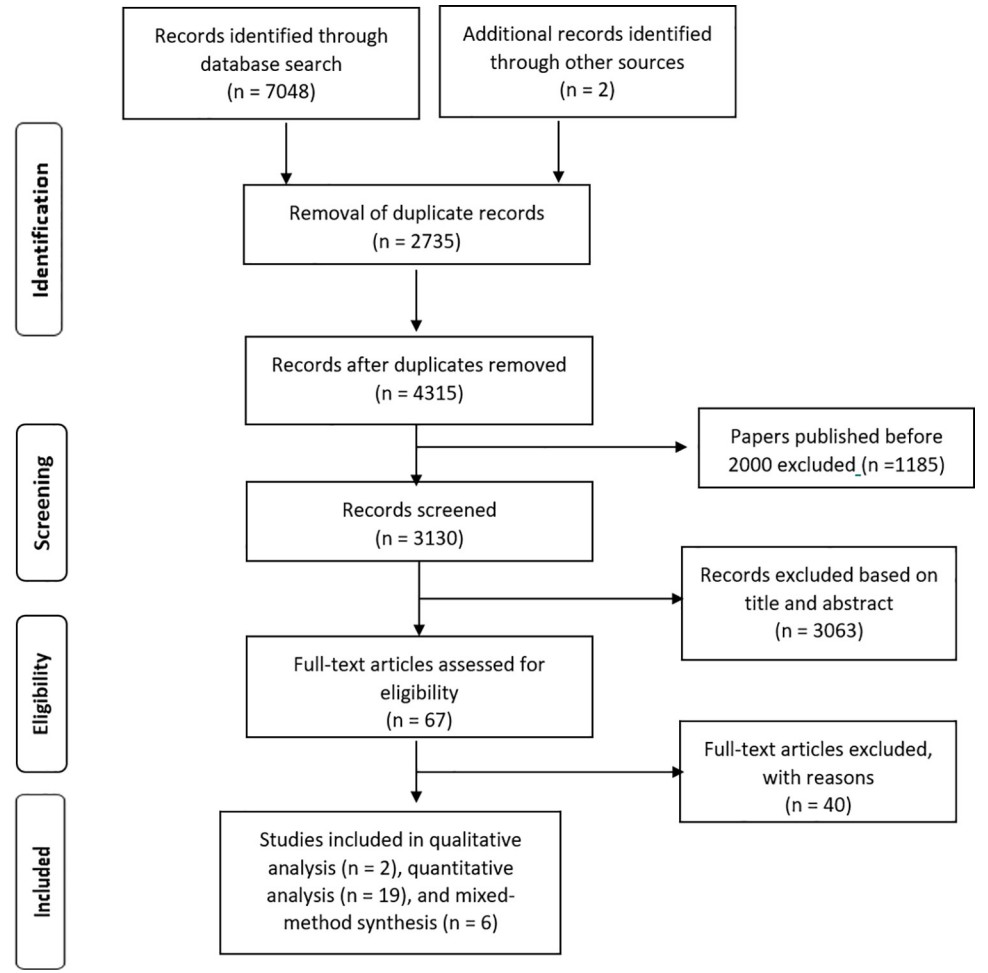

**Fig 1. Flow diagram of paper selection process.**

43.2%. The scatter plots show a linear correlation between the delayed period (in mean and median values) and G2D (Fig 3).

## Health-service-seeking behavior

A relationship between delayed detection and health-service-seeking behavior was reported by 14 of the 15 studies that assessed such an association (92.3%). Statistically significant risk factors for detection delay were found for the following: visiting traditional or alternative medicine suppliers, medicine shops, and private healthcare as the first point of care; and taking no action after the appearance of signs and symptoms. ORs for these risk factors ranged from 2.6 to 10.4 [25,27,30,32].

One study found that many leprosy patients (33/47; 70.2%) took no care-seeking action after noticing the first sign of leprosy [20]. In four other studies, 12–59% of patients started seeking care by buying medicine at a medicine shop or pharmacy, or by visiting a traditional healer [30,33,34,39]. In one of these four studies [30], a quarter of the patients started seeking care by visiting a private doctor or clinic. Finally, another study described that people with leprosy initially did not seek (as the authors termed) 'appropriate' health care [37].

Qualitative studies indicated that some patients perceived visiting a doctor and spending time and money on such a visit as "a waste". However, a preference for seeking care from a

**Table 1. Study characteristics of the included papers.**

| First Author, Year | Study design | Country | Setting | Sample Size (Response Rate) |
|---|---|---|---|---|
| **Quantitative Study Design** | | | | |
| **Libardo Gomez, 2018** [24] | Observational; by interview with cross-sectional study | Colombia | Community | 249 |
| **Peter G Nicholls, 2005** [25] | Cross-sectional with structured interviews | India | Community | 356 |
| **Peter G Nicholls, 2003** [26] | Observational; by patient cohort | Bangladesh, India | Hospital | 2664 |
| **A Samraj, 2012** [20] | Observational; descriptive cross-sectional with interview | India | Hospital | 86 |
| **Mary Henry, 2016** [27] | Observational; explorative study with a quantitative questionnaire (cross-sectional) | Brazil | Clinic | 122 |
| **Natasja van Veen, 2007** [14] | Observational; long-term prospective cohort study | Bangladesh, Ethiopia | Clinic, Community | Total: 3250 (1594; 49%) Ethiopia: 586 (517; 88.2%) and Bangladesh: 2664 (1077; 40.4%) |
| **Furen Zhang, 2009** [28] | Observational; by interview (cross-sectional) | China | Community | 88 |
| **XS Chen, 2000** [29] | Cross-sectional design with patients' records | China | National | 27,928 |
| **Linda M Robertson, 2000**] [30] | Cross-sectional with a structured questionnaire | Nepal | Hospital | 166 |
| **DNJ Lockwood, 2001** [31] | Cross-sectional design with case-note review | UK | Hospital | 28 |
| **Patricia D. Deps, 2006** [15] | Cross-sectional; descriptive with interviews | Brazil | Community | 506 (450; 88.9%) |
| **Cacilda Da Silva Souza, 2003** [16] | Cross-sectional with semi-structured interviews | Brazil | Hospital, Clinic | 40 |
| **Tigist Shumet, 2015** [17] | Observational; cross-sectional retrospective record review | Ethiopia | Hospital | 513 |
| **Govindarijulu Srinivas, 2019** [13] | Observational; case-control study with interview | India | Community | 280 |
| **Jin Lan Li, 2016** [18] | Observational; by patients' records (cross-sectional) | China | Community | 1274 |
| **Mahdis Ghavidel,2018** [19] | Observational; cross-sectional study | Iran | Clinic | 42 |
| **Marcos Tu´lio Raposo,2018** [21] | Observational; cross-sectional study | Brazil | Community | 249 (222; 89.2%) |
| **Sabeena J,2020** [22] | Observational; retrospective | India | Hospital | 403 |
| **Tongsheng Chu, 2020** [23] | Observational; retrospective | China | Community | 232 |
| **Qualitative Study Design** | | | | |
| **Peter G Nicholls, 2003** [38] | Participatory method with semi-structured interviews, focus groups, observation, and free listing | Paraguay | Hospital | 36 |
| **Carmen Osorio-Mejia,2020** [39] | Qualitative method with semi-structured interviews | Peru | Clinic | 30 |
| **Mix Method Design** | | | | |
| **Thirumugam Muthuvel, 2017** [32] | Quantitative component with a matched case-control design with interviews, followed by a descriptive qualitative component | India | Community | 210 |
| **Sonia F. Raffe, 2013** [34] | Quantitative component with a cross-sectional approach. Qualitative data were collected from semi-structured interviews with patients, case-notes review, and brief clinical examinations | Nepal | Hospital | 78 (75; 96.2%) |
| **Ulla Britt Engelbrektsson,2019** [35] | Quantitative and qualitative method with interview and review of patient's documents | Nepal | Hospital | 81 |
| **Sachin Ramchandra Atre, 2011** [33] | Cross-sectional descriptive and qualitative design with semi-structured interviews | India | Community | 58 |
| **Cavalcante MDMA, 2020** [36] | The quantitative data on the notified cases were provided by the program's municipal coordinator, and the qualitative data were obtained by semi-structured script | Brazil | Clinic | 19 |

*(Continued)*

**Table 1.** (Continued)

| First Author, Year | Study design | Country | Setting | Sample Size (Response Rate) |
|---|---|---|---|---|
| **Myo Ko Ko Zaw, 2020** [37] | The quantitative analysis used an ecological study design, and the qualitative data were collected by interview | Myanmar | Clinic, Region Aggregate Data | 42 |

traditional healer resulted in delayed case detection [26,32]. Three studies reported that the delay could have been reduced by visiting a clinic that was nearest to the patients' house (OR = 0.24; 95% CI = 0.27–0.70), by seeking care immediately after noticing the first symptom (p = 0.017), or if they had had better access to health service (p<0.01) [24,25,29].

Fig 4 summarizes the pathway from health-service seeking to leprosy diagnosis. The pathway shows three possible levels of care for which leprosy patients sought diagnosis, and also indicates the flow of health-service seeking. As patients may not notice the initial appearance of leprosy signs or symptoms, they may take no action to seek care. When they do notice these signs or symptoms, they may: (a) still ignore them and take no action; (b) take self-medication; or (c) visit healthcare providers. The three levels of care shown in this figure indicate the type of facilities at which a diagnosis can be established. Level 1 refers to seeking care from non-formal healthcare providers (i.e., medicine and home remedy shop, non-qualified practitioner (a health practitioner who does not have official training for diagnosing leprosy), or traditional medicine), where leprosy cannot be diagnosed or is potentially missed. Level 2 refers to health-service facilities at which leprosy can be diagnosed and patients can start multidrug therapy (MDT). Facilities at this level include medical doctors, clinics, local health posts, private health services, public health services, and hospitals. However, at this level, too, it is possible that leprosy is not diagnosed, and that several more visits are needed before a diagnosis is made and MDT can start. Level 3 refers to specialist leprosy services or referral hospitals. At this level, leprosy can be diagnosed, and patients can start MDT immediately.

The color of the arrows on the pathway chart indicates three steps in care-seeking: blue for the first visit, red for the second and subsequent visits, and black for active case detection by health staff. The chart shows that someone may need to make a series of visits before they are diagnosed, a possibility that is due partly to leprosy's susceptibility to misdiagnosis, which can therefore lead to several visits, and possibly referrals to other health services.

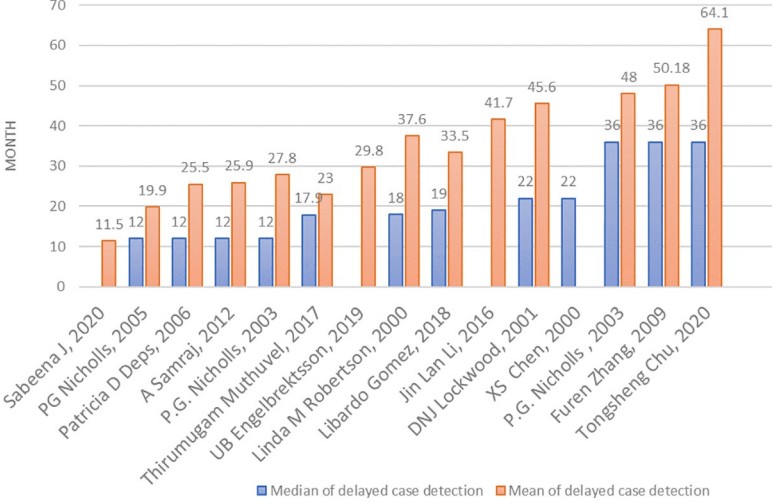

**Fig 2. The median and mean time of delayed case detection, in months.**

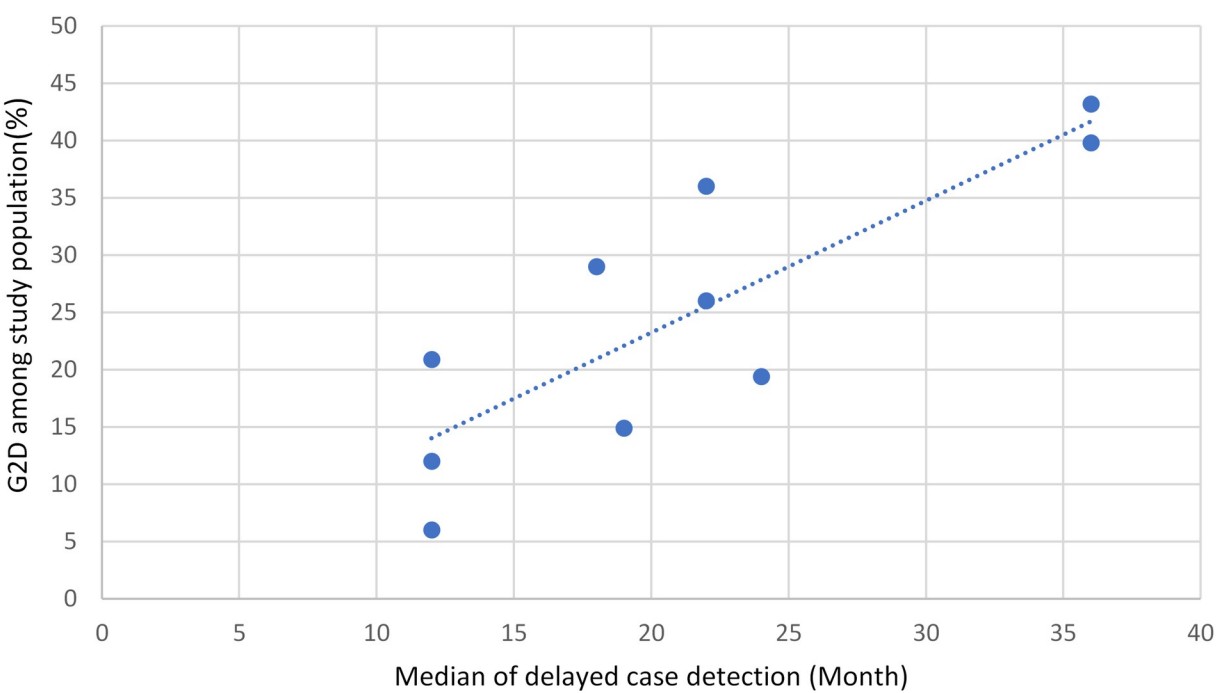

(a)

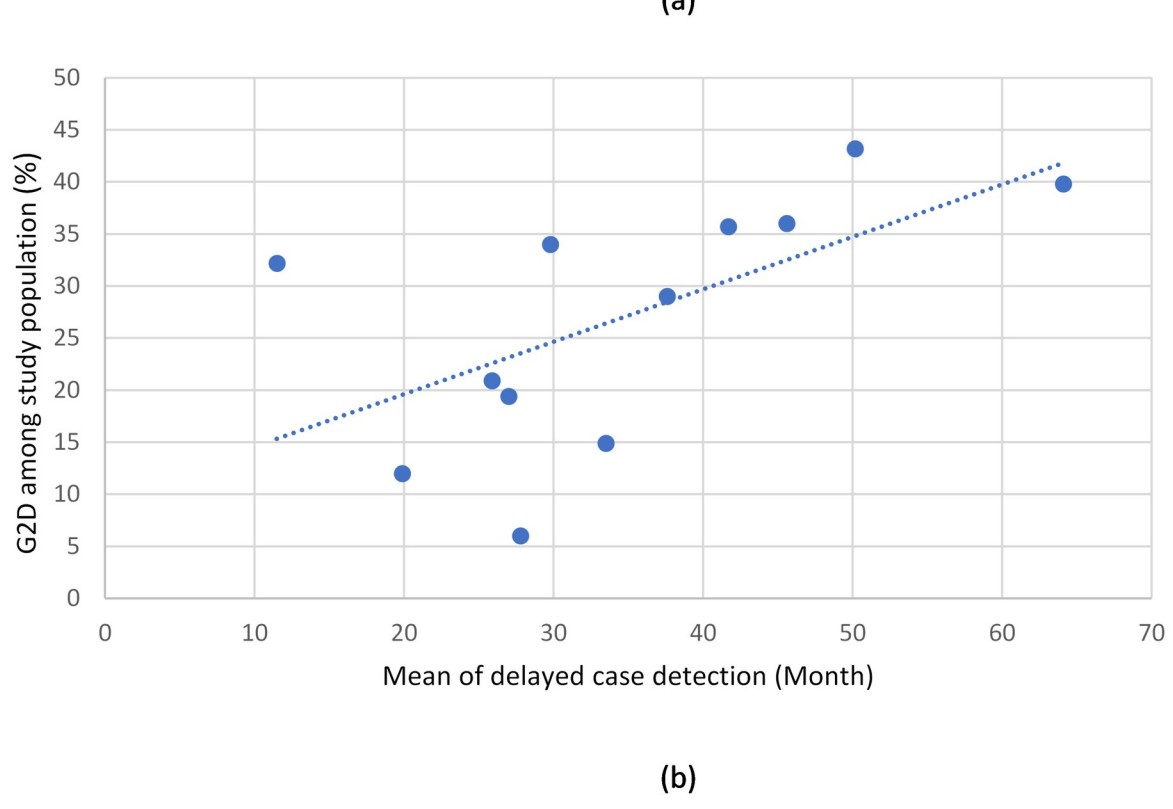

(b)

**Fig 3. Correlation between (a) median time of delayed case detection and percentage of G2D and (b) mean time of delayed case detection and percentage of G2D.**

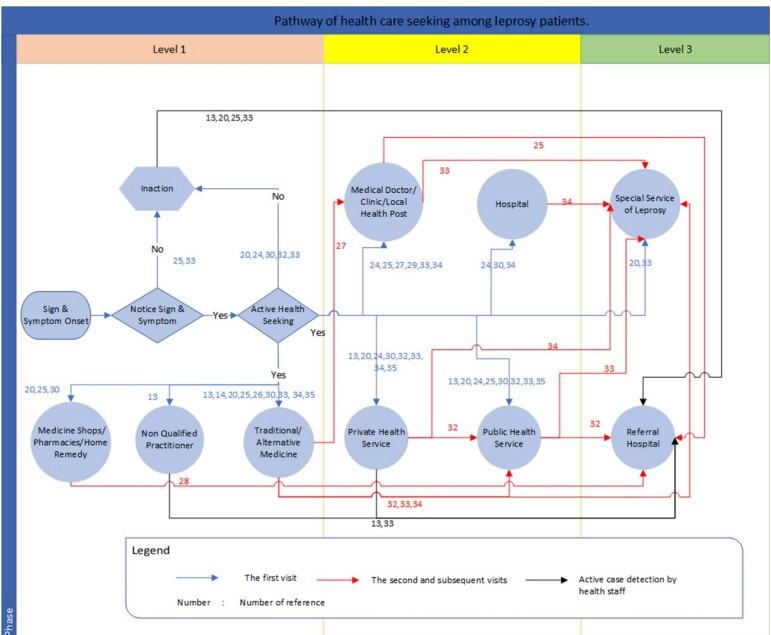

**Fig 4. Pathway of health care seeking among leprosy patients.**

## Individual factors

**Age.** Sixteen studies investigated the association between age and detection delay [13–15,17,18,20–30]. Ten of them (62.5%) reported a statistically significant association [13,14,17,18,21,22,25,26,29,30]. Three of the nine studies reported that an age of 50 years and above was a risk factor for detection delay, the respective ORs being 6.6, 3.52, and 2.2 [13,14,17]. Other studies reported that the risk factors for detection delay were highest among patients aged 30 years and above: 30–44 years (OR = 2.12) [14]; 36 years and above (OR = 2.03) [25]; 31–60 years (OR = 1.2) [13]; 45–59 years (OR = 3.44) [14], and above 45 years (OR = 2.12) [22]. Two studies did not report ORs, but indicated significant associations with delay in the age group >65 years [26], and in the age group >15 years [18]. One study reported that delay percentages were highest in the 45–54 year and the 55–64 year age groups [29]. One study reported that the mean age of individuals with G2D was significantly higher than that of individuals without G2D (p<0.008)[21].

**Sex.** Three [18,19,22] of the ten studies investigating the role of sex [14,18–22,25,27,28,33] reported a lower incidence of G2D among females than males, and two [14,27] reported shorter delay in detection among females.

**Type of leprosy.** Eight [13,15,21,22,25,28,29,31] of eleven studies that investigated type of leprosy [13,15,19,21–25,28,29,31] reported a significantly longer detection delay in patients with multibacillary (MB) leprosy than in those with paucibacillary (PB) leprosy. ORs ranged from 1.8 to 9.1. One study reported longer detection delay in patients with PB leprosy (OR = 2.76) [23].

**Symptom perception.** Eight [13,25,27,28,32,35,37,38] of the nine studies investigating symptom perception [13,20,25,27,28,32,35,37,38] reported an association between symptom perception and delayed case detection. Two of the eight reported a statistically significant association with detection delay [27,28]. Most patients either did not know the signs or symptoms of leprosy, and therefore ignored them; or, even if they noticed them, thought they would

disappear spontaneously [13,20,32,38]. Some qualitative studies reported that patients and families recognized leprosy only when symptoms of ulcers, deformity, or wounds were advanced, and that a lack of concern about initial symptoms contributed to the delay in detection [32,35,37,38].

**Knowledge.** All ten studies that investigated knowledge reported that it was possibly associated with delayed detection [16,25,32–39]. One of these studies reported that not knowing the cause of leprosy was a significant risk factor for delay (OR = 1.89)[25]. Eight other studies reported that unawareness of leprosy and a lack of knowledge about it were due to the lower priority given to health than to wage-earning [16,32,34–39]. This meant that, as long as it was not painful, leprosy was considered not to be important [38].

**Other individual factors.** Three other factors were also stated to be associated with delayed case detection: sharing a house with a person affected by leprosy; the walking-time to a health service; and alcohol consumption [30,32].

## Socioeconomic factors

**Location.** Three [25,30,33] of the seven studies that investigated the role of residence [21,24,25,27,30,32,33] reported an association with delayed case detection. Rural residence was a statistically significant indicator of delay (ORs ranged between 0.47 to 0.59) [25,30]. Long distances to health services were also associated with delay][33].

**Educational level and occupation.** One [30] of the eight studies [20,21,23–25,28,30,32] that investigated the role of educational status reported a possible association with delayed case detection (OR = 2.1). Two studies reported that working with daily wage labor or in agricultural sector (OR = 1.5; 95% CI = 1.1–2.2) or being unemployed (OR = 7.70; 95% CI = 2.88–20.6), can increase the risk of delay [13,21]. One study reported a significant association between occupation and detection delay (p<0.01), with farmers having the longest delay [29].

**Other socioeconomic factors.** Two studies from China reported ethnic group or nationality as a risk factor for delayed case detection[18,29]. Four studies found that delayed case detection was not associated with income, health insurance schemes, or marital status [21,24,27,32].

## Social and community factors

**Stigma.** Six studies investigated the role of stigma [27,32,33,36–38]. One of these, a quantitative study [27], reported a significant association between delay and the fear of isolation (OR = 10.37; 95% CI = 2.2–49.5). A qualitative study [38] reported that stigmatization was reinforced by isolation policy, church teaching, a belief that leprosy is highly contagious, fear of leprosy, leprosy being a taboo subject, and references to leprosy as "a disease of society, not of people". Another of the six studies that investigated stigma [33] reported that over two-thirds of patients did not disclose their condition to their community. Two studies with a mixed method approach reported that there is still a stigma in leprosy and that people are afraid that if their disease becomes known, this will cause discrimination and stigmatization by family and community [36,37].

**Awareness and beliefs.** One [25] of four studies that investigated social values [25,32,37,38] found that shorter delay was associated with a belief that leprosy was caused by a curse, spirit, or ghost (OR = 0.28; 95% CI = 0.08–0.97]), but one other study indicated that these kinds of beliefs led to longer delay [38]. Another of these studies reported that high trust in a traditional healer [38] and lack of social awareness [32] were major contributors to delay. In one study, it was observed that the interest in and social awareness of leprosy reduced after reaching elimination of leprosy as a public health problem [37].

**Geographic area.** One [29] of three studies investigating the role of geographic area [27,29,31] reported a delay in case detection that was shorter (21 months) in a non-endemic area than in an endemic area (23 months) (p<0.01).

## Discussion

One highlight of this systematic review is that leprosy case detection is often delayed, with median delay ranging from 12 to 36 months, and mean delay from 20 to 50 months. The percentage of grade 2 disability (G2D) ranged from 5.6% to 43.2%. A linear correlation between delayed case detection and the percentage of G2D indicated that the longer the delay, the more common and the greater the severity. The most prominent factor associated with delayed case detection was health-service-seeking behavior. Individual factors associated with delayed case detection were older age, being male, having a lower perception of disease symptoms, having MB leprosy, and a lack of knowledge. The most identified socioeconomic factors associated with delayed case detection were living in a rural area, performing labor for a daily wage labor–including agricultural labor–and being unemployed. The most reported social and community factor associated with delayed case detection was stigma.

A highlight of this review is that delayed case detection is closely related to health-service-seeking behavior, i.e., seeking care from qualified healthcare facilities in a timely way. Health-service seeking is a complicated issue, as it involves a complex paradigm of social, historical, cultural, and economic variables, all of which define a person's mindset [40]. There are several reasons why people with leprosy may not seek care: stigmatization, social values, poor knowledge of leprosy signs and symptoms, and poor access to healthcare services [40]. The extent to which people are able to correctly interpret the early or later symptoms of leprosy is associated with their level of knowledge. It will also influence their health-service-seeking behavior [40]. People who misinterpret their symptoms or do not recognize them are more likely to ignore the first signs of their disease and thus take no action [20,25,27,32,38]. Like stigma and social values, beliefs that leprosy is caused by a curse or a spirit or other supernatural cause may also cause people who have early signs of leprosy not to seek timely treatment at qualified healthcare services, but to take self-medication, visit non-qualified practitioners of traditional or popular medicine, or visit a medicine shop [25,38,40,41]. Stigma and the fear of it can lead people with leprosy to conceal their condition, or to visit a distant health center in order to avoid being recognized by people from their community, and could thus cause delayed case detection [33,40–42]. Being male, being older, having a poor knowledge of leprosy, lacking perception of the initial symptoms, and being unaware of the severity of symptoms are all associated with inadequate decisions about seeking treatment from the health services [40,43–45]. To reduce the delay to a minimum, people need to recognize the severity of early symptoms, seek care as soon as possible after noticing possible symptoms, and avoid multiple visits to inadequate care providers by visiting qualified ones.

Inadequate health-service-seeking behavior by people with leprosy is also affected by socioeconomic factors [40,46]. Before they finally visit qualified healthcare services, many people from poor households who contract leprosy first take self-medication or visit a traditional healer [45,47]. As the symptoms of leprosy often appear without causing pain, they tend not to be perceived as a physical health problem–thereby providing another reason for people to delay seeking healthcare on the grounds that it would waste time and money [32]. Case detection is often delayed more by unemployed people and laborers on a daily wage than by factory workers, office workers, and students [13,29]. In contrast, better health-service seeking by people living in rural areas is associated with a higher monthly income and with living close to

health services [48]. To improve healthcare seeking behavior, it is therefore important specifically to target people who live in rural areas and those working as laborers on a daily wage.

This review also underlines the strong relationship between delayed case detection and the risk of disability in leprosy patients [14,28,49]. As an important complication of leprosy, disability has a strong and often life-long impact on the person affected by the disease. It can be prevented by early detection and adequate treatment, thereby contributing considerably to reductions in disease burden [7]. If not prevented it can also become part of a vicious circle: because patients with visible disabilities and ulcers on their hands and feet often face stigmatization, they may postpone help-seeking, thus further delaying detection [50]. Even if the leprosy infection is cured, leprosy patients may have lasting physical and mental disabilities, and continue to face stigma, discrimination, and social exclusion [51–53].

Based on the healthcare-seeking pathway we derived from this systematic review (Fig 4), interventions to improve leprosy knowledge, awareness, and perception will play a crucial role in reducing detection delays. Health education is the most common intervention, both for bridging gaps in information and knowledge, and also for promoting early detection [54]. Helping people to recognize leprosy symptoms on time could improve earlier care seeking [55]. As people may also worry about stigmatization and income loss after being diagnosed [56], health education should focus on convincing them that receiving appropriate treatment is the best option not only for their health and future income, but also for avoiding the stigma related to the disease [38,57].

Health education requires the following: well-designed programs; good materials developed on the basis of a strong methodology; information and messages that are sensitive to local culture; and appropriate targeting strategies to groups and individuals in the community. Such strategies should include adult literacy programs [54,58]. The interventions should also be tailored to specific priority subgroups in the population: the elderly, males, manual laborers and the unemployed; males in rural areas, especially those at a long distance from health services; people in endemic areas where there is a high prevalence of MB leprosy; and communities characterized by high levels of stigma and by social values that prefer traditional medicine [16,30].

One promising strategy in leprosy-related health education is to invite former leprosy patients to become health educators in their community–the so-called "contact intervention" strategy, which is both effective and replicable [59]. A range of methods is also available for delivering education through Information Education and Communication (IEC) campaigns: these include TV, radio, posters, pamphlets, IEC vans, film shows, and folk dances [60]. Other strategies for which there is some evidence of effectiveness include the integration of leprosy programs into general healthcare, and IEC programs that use socioeconomic rehabilitation to reduce the stigma of leprosy in the community [61].

Healthcare-seeking behavior among leprosy patients (Fig 4) can follow a pathway comparable to that of tuberculosis (TB) patients [62]. Before finally being diagnosed, patients may visit several health facilities, drug stores or traditional medicine practitioners [63,64]. Leprosy patients often visit private health facilities as their first point of care, which are sometimes known to result in diagnosis delay, higher incurred costs, and more severe disability [65]. To reduce diagnostic delay, a national leprosy program can adopt approaches developed by the TB program, such as the public-private mixed (PPM) approach and an approach involving community health workers in the national leprosy program [66,67].

This is the first systematic review on individual and socioeconomic factors related to case detection in leprosy. Not only does it summarize the complex health-seeking behavior of leprosy patients in a simple figure showing how care seeking-behavior is related to detection delay, it also confirms the strong linear correlation between delay and G2D.

The study also has some limitations. First, because methods and research settings varied between studies, it is difficult to generalize our findings. Second, as we included only literature in English, we could not capture publications in other languages that may have originated in countries with a high leprosy burden, such as Indonesia and Brazil [9]. Third, as the studies used various definitions and cut-off periods for detection delay, standardization was difficult. In our view, a uniform definition of case detection delay in leprosy is therefore required for future policy development, for which we propose six months or one year as a threshold [14,26,28]. Fourth, the WHO leprosy disability grading system grades patients according to the presence of disabilities of the eyes, hands, and feet [4,68]. G2D is usually reported to WHO as the proportion of people with G2D at any body site among leprosy cases newly diagnosed in a specific year. The sum score of these six body sites is called the Eye-Hand-Foot (EHF) score and is used as an overall indicator of the impairment status of an individual with leprosy [68]. Taking EHF scores into account instead of G2D could provide a more nuanced insight into the correlation between detection delay and level of disability. Unfortunately, EHF scores are not widely available. Finally, our review describes and identifies only the demand-side factors of detection delay–the individual and social factors–that affect detection delay; it has not captured the supply-side factors, i.e., those involving health services. To provide a comprehensive picture of case detection delay in leprosy, these also need to be studied.

## Conclusion

This review confirms that delayed case detection is clearly correlated with increased disability in leprosy, and therefore the reduction of detection delay should be a priority of leprosy programs. Interventions should focus on health-service-seeking behavior, and should consider relevant individual, socioeconomic, and community factors, including stigmatization. To increase knowledge and perceptions of initial symptoms, health education should target high-risk groups. For a comprehensive understanding of factors associated with case detection delay in leprosy, further study is required of health-service-related factors that contribute to delayed detection.

## Supporting information

**S1 Table. PRISMA Checklist.**
(DOC)

**S1 Text. Search strategy for each database.**
(DOCX)

## Acknowledgments

The authors also thank Wichor Bramer, Sabrina Meertens-Gunput, Elise Krabbendam, Maarten Engel, and Christa Niehot from the Erasmus MC Medical Library for developing and updating the search strategies. We also thank to Aswath Karanukaran, Anna van 't Noordende, and Jiske Erlings for helping to collect the studies included in this systematic review.

## Author Contributions

**Conceptualization:** Yudhy Dharmawan, Ida Korfage, Jan Hendrik Richardus.

**Data curation:** Yudhy Dharmawan.

**Formal analysis:** Yudhy Dharmawan, Ahmad Fuady.

**Investigation:** Yudhy Dharmawan.

**Methodology:** Yudhy Dharmawan, Ida Korfage, Jan Hendrik Richardus.

**Supervision:** Ahmad Fuady, Ida Korfage, Jan Hendrik Richardus.

**Validation:** Yudhy Dharmawan, Ahmad Fuady, Ida Korfage, Jan Hendrik Richardus.

**Visualization:** Yudhy Dharmawan.

**Writing – original draft:** Yudhy Dharmawan.

**Writing – review & editing:** Yudhy Dharmawan, Ahmad Fuady, Ida Korfage, Jan Hendrik Richardus.

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
