## [Decision Letter · Decision Letter 0]

27 May 2021

Dear Mr Dharmawan,

Thank you very much for submitting your manuscript "Individual and community factors determining delayed leprosy case detection: A systematic review" for consideration at PLOS Neglected Tropical Diseases. As with all papers reviewed by the journal, your manuscript was reviewed by members of the editorial board and by several independent reviewers. In light of the reviews (below this email), we would like to invite the resubmission of a significantly-revised version that takes into account the reviewers' comments. 

We cannot make any decision about publication until we have seen the revised manuscript and your response to the reviewers' comments. Your revised manuscript is also likely to be sent to reviewers for further evaluation.

Sincerely,

Alberto Novaes Ramos Jr

Associate Editor

Kristien Verdonck

Deputy Editor

Reviewer's Responses to Questions

**Key Review Criteria Required for Acceptance?**

**Methods**

-Are the objectives of the study clearly articulated with a clear testable hypothesis stated?

-Is the study design appropriate to address the stated objectives?

-Is the population clearly described and appropriate for the hypothesis being tested?

-Is the sample size sufficient to ensure adequate power to address the hypothesis being tested?

-Were correct statistical analysis used to support conclusions?

-Are there concerns about ethical or regulatory requirements being met?

Reviewer #1: Proceed without modifications

Reviewer #2: 1. The objectives of the study are clearly articulated with a clear testable hypothesis stated. 

2. The study design is appropriate to address the stated objectives.

3. Since the study is a systematic review about individual and community factors determining delayed leprosy case detection, the population, sample size, statistical analysis are not involved in the study. 

4. The methods for systematic review in the study is proper.

Reviewer #3: In the background the authors mentioned the increased chance of disability by delayed case detection. At the same time, they used physical disability as a measure of delay and analyzed this factor (G2). In the conclusion, they affirmed “Delayed leprosy case detection is clearly correlated with increased disability”. To maintain this conclusion the aim has to be changed to confirm this issue.

See in Method: Repeated - To select articles for full-text screening, two reviewers (YD and AF) independently screened article titles and abstracts. Data from articles were extracted and double-entered into Microsoft Excel: two reviewers (YD and AF) independently screened article titles and abstracts and then extracted  the selected articles and double-entered these into Microsoft Excel.

**Results**

-Does the analysis presented match the analysis plan?

-Are the results clearly and completely presented?

-Are the figures (Tables, Images) of sufficient quality for clarity?

Reviewer #1: Please find all comments and recommendations as an attached document

Reviewer #2: The analysis presented matches the analysis plan and the results are clearly and completely presented.

**Conclusions**

-Are the conclusions supported by the data presented?

-Are the limitations of analysis clearly described?

-Do the authors discuss how these data can be helpful to advance our understanding of the topic under study?

-Is public health relevance addressed?

Reviewer #1: Please find all comments and recommendations as an attached document

Reviewer #2: The conclusions are supported by the data presented.

The limitations of analysis are clearly described.

Public health relevance is addressed.

Reviewer #3: The aim is “To identify determinants of delayed detection or Grade 2 Disability (G2D),” It means that the Grade 2 Disability is assumed as the same (the expression) of delayed detection. This way, it cannot be the conclusion.

**Editorial and Data Presentation Modifications?**

Reviewer #1: Please find all comments and recommendations as an attached document

Reviewer #3: The authors mentioned the limitations for the use of only publications in English. However, they do not mention the reasons why the results of Leprosy Monitoring Evaluation (LEM) usually publicized in English were not used. In the LEM reports the time between the sings and symptoms until the diagnosed is one of the most important point of the analyses. The results usually are the opposite of the findings: “ One [29] of three studies investigating the role of geographic area [27, 29, 31] reported a delay in  case detection that was shorter (21 months) in a non-endemic area than in an endemic area (23  months) (p<0.01).”

Usually the less endemic more delay of diagnosis explained by the centralization of services of leprosy diagnosis - it is the rational for rare diseases when the doctors are not prepared to identify the signs and symptoms. 

Some individual and community factors were explained by the stigmatization of the disease. The stigma is known a problem for leprosy control. However, the social vulnerability resulted by social exclusion, poverty, illiteracy, difficulties to access of health services are different of stigma. Considering leprosy as a neglected tropical disease the authors should discuss this problem independently of stigma.

**Summary and General Comments**

Reviewer #1: Please find all comments and recommendations as an attached document

Reviewer #2: If all factors related to individual and community were presented in one table or several figures, it would be easy to read for readers since it only described the reviewed papers.

Reviewer #3: The absence of the LEM results carried out in the endemic countries for many rounds should be justified.

PLOS authors have the option to publish the peer review history of their article (what does this mean?). If published, this will include your full peer review and any attached files.

Reviewer #1: Yes: Mauricio Lisboa Nobre

Reviewer #2: No

Reviewer #3: No
---

## [Decision Letter · Decision Letter 1]

14 Jul 2021

Dear Mr Dharmawan,

We are pleased to inform you that your manuscript 'Individual and community factors determining delayed leprosy case detection: A systematic review' has been provisionally accepted for publication in PLOS Neglected Tropical Diseases.

Best regards,

Alberto Novaes Ramos Jr

Associate Editor

Kristien Verdonck

Deputy Editor

Reviewer's Responses to Questions

**Key Review Criteria Required for Acceptance?**

**Methods**

-Are the objectives of the study clearly articulated with a clear testable hypothesis stated?

-Is the study design appropriate to address the stated objectives?

-Is the population clearly described and appropriate for the hypothesis being tested?

-Is the sample size sufficient to ensure adequate power to address the hypothesis being tested?

-Were correct statistical analysis used to support conclusions?

-Are there concerns about ethical or regulatory requirements being met?

Reviewer #2: (No Response)

Reviewer #3: (No Response)

**Results**

-Does the analysis presented match the analysis plan?

-Are the results clearly and completely presented?

-Are the figures (Tables, Images) of sufficient quality for clarity?

Reviewer #2: (No Response)

Reviewer #3: (No Response)

**Conclusions**

-Are the conclusions supported by the data presented?

-Are the limitations of analysis clearly described?

-Do the authors discuss how these data can be helpful to advance our understanding of the topic under study?

-Is public health relevance addressed?

Reviewer #2: (No Response)

Reviewer #3: (No Response)

**Editorial and Data Presentation Modifications?**

Reviewer #2: (No Response)

Reviewer #3: (No Response)

**Summary and General Comments**

Reviewer #2: (No Response)

Reviewer #3: (No Response)

PLOS authors have the option to publish the peer review history of their article (what does this mean?). If published, this will include your full peer review and any attached files.

Reviewer #2: No

Reviewer #3: **Yes: **Eliane Ignotti

---

## [Editor Report · Acceptance letter]

29 Jul 2021

Dear Mr Dharmawan,

We are delighted to inform you that your manuscript, "Individual and community factors determining delayed leprosy case detection: A systematic review," has been formally accepted for publication in PLOS Neglected Tropical Diseases.

Best regards,

Shaden Kamhawi

co-Editor-in-Chief

Paul Brindley

co-Editor-in-Chief
